



# Ozonolysis of primary biomass burning organic aerosol particles: Insights into reactivity and phase state

Sophie Bogler[1], Jun Zhang[1], Rico K. Y. Cheung[1], Kun Li[1,2], André S. H. Prévôt[1], Imad El Haddad[1], David M. Bell[1*]

[1]Paul Scherrer Institute, Center for Energy and Environmental Sciences, Villigen, 5232, Switzerland

[2]Qingdao Key Laboratory for Prevention and Control of Atmospheric Pollution in Coastal Cities, Environment Research Institute, Shandong University, Qingdao 266237, China

*Correspondence to*: david.bell@psi.ch

**Abstract.** Biomass burning organic aerosol (BBOA) particles are a major contributor to atmospheric particulate matter with various effects on climate and public health. Quantifying these effects is limited by our understanding of the BBOA particles' evolving chemical composition during atmospheric aging, driven by their exposure to atmospheric oxidants. This study explores the role of ozone ($O_3$) as atmospheric oxidant in processing primary BBOA particles. We exposed particulate emissions from beech, spruce and pine wood fires to $O_3$ in an oxidative flow reactor, monitoring their chemical evolution using high-resolution time-of-flight aerosol mass spectrometry (HR-AMS) and extractive electrospray ionization time-of-flight mass spectrometry (EESI-ToF). We found that the oxidative state of the particles increased with $O_3$ exposure, as shown by the consistent, albeit minor, rise in O/C ratios. Analysis of the EESI-ToF data revealed specific molecular groups containing 18 and 20 carbon atoms, likely mainly abietic, linoleic, and oleic acid, as highly reactive toward $O_3$ and driving the increase in oxidative state. At higher relative humidity, increased oxidation and loss of reactive species indicate that enhanced $O_3$ diffusion into particles allows the ozonolysis to progress further, highlighting humidity's role in overcoming diffusion barriers that limit the ozonolysis in dry conditions. This study provides qualitative insights into the oxidative processing of primary BBOA particles in different phase states, presenting $O_3$ as selective oxidant. Further research could focus on quantifying the progression of the ozonolysis, in particular the change in diffusion rates depending on relative humidity conditions or particle sizes.

## 1 Introduction

Atmospheric particulate matter (PM) affects our climate and public health. The importance of PM for human health is based on damaging effects on the respiratory and cardiovascular system after extended exposure to atmospheric PM. For the year 2015, an estimated 8.9 million of premature deaths worldwide were linked to this exposure (Burnett et al., 2018), and air pollution was listed in the top five global mortality risk factors (Lelieveld et al., 2015). Given this ranking, it is evident that advancing the understanding of toxic PM sources and their mitigation is crucial to reduce future health impacts. One key



contributor to atmospheric submicron PM are organic aerosol (OA) particles, which account for up to 90 % of PM mass (Jimenez et al., 2009).

One of the main sources of primary OA particles are combustion processes such as vehicular exhaust, and biomass burning. Biomass-burning OA (BBOA) makes up 15-68 % of the total OA mass estimated in Europe during winter in rural areas
(Puxbaum et al., 2007) when most of the BBOA is generated for residential heating. During summer, BBOA comes predominately from prescribed burnings and wildfires. Both fire suppression and climate change have resulted in an increase in the frequency of wildfires, and thus increasing the BBOA contribution to atmospheric PM (Iglesias et al., 2022; Westerling, 2016).

To assess the impacts of atmospheric BBOA, it is required to characterize the chemical composition and its evolution through various atmospheric aging pathways. As an analytical method, aerosol mass spectrometry has been used to characterize BBOA either offline (e.g. Daellenbach et al., 2016; Liang et al., 2022) or online, in the field (e.g. Saarnio et al., 2013; Hayden et al., 2022) or in laboratory experiments (e.g. Yazdani et al., 2021). Still, resolving BBOA's chemical composition remains an on-going analytical challenge, given the presence of millions of individual species (Zhang et al., 2023). Each compound has
unique physicochemical properties, which determine its reactivity towards different atmospheric oxidants, its partitioning behaviour, both of which control its atmospheric lifetime.

When exposed to sunlight and to atmospheric oxidants, such as the hydroxyl radical (OH), ozone ($O_3$) and the nitrate radical, the chemical composition of the primary BBOA particles evolves in the atmosphere to form aged BBOA. Aged BBOA
generally has a higher oxidation state and has lost mass due to evaporation (Yazdani et al., 2021). Given the importance of these atmospheric processing pathways, knowing only the chemical composition of primary BBOA and a few marker species is not sufficient to assess the overall impact of BBOA during their atmospheric lifetime. Also, the use of markers as efficient tracers is limited depending on their stability towards the various oxidants. For example, several studies have shown how levoglucosan decays by oxidation with OH over the course of hours (Bertrand et al., 2018b; Hennigan et al., 2010; Hoffmann
et al., 2010; Lai et al., 2014).

Laboratory studies targeting specific aging mechanism with high chemical detail remain scarce. First results from Fortenberry et al., 2018 show how photochemical aging depletes primary BBOA emissions overall, while simultaneously enhancing certain marker signals that are likely produced through secondary oxidation processes. In a similar experimental setup, Bertrand et al.,
2018 highlighted how the majority of primary markers from woodstove emissions are lost through aging by OH. Most secondary compounds with potential use as tracers remain yet unidentified. Yazdani et al., 2021 used mid-infrared spectroscopy to analyse the functional group composition in BBOA aged with OH or nitrate radicals compared to freshly




emitted BBOA from wood burning. Here, the authors illustrate the importance of oxygenated functional groups like carboxylic acids in aged compounds, while the backbone of the parent compound structure is retained.


One of the ubiquitous oxidants in the atmosphere is $O_3$, where $O_3$ reacts predominately with alkenes (J. Ziemann, 2005) (Zelenyuk et al., 2017). Previous work showed limited changes in the oxygen-to-carbon ratio (O:C) of BBOA upon $O_3$ exposure as measured by an aerosol mass spectrometer (AMS) (Browne et al., 2019). However, it is unknown whether these limited changes with $O_3$ exposure were due to diffusion limitations or the limited reactivity of BBOA and whether BBOA

reactivity depends on its chemical composition, hence the type of fuel burnt and combustion conditions. These experimental findings were also limited by the chemical resolution of the AMS. Because $O_3$ is thought to react predominately with alkenes, many studies have probed the reactions of alkenes in the aerosol with $O_3$. A common alkene studied is oleic acid, and aging in the presence of $O_3$ has shown the reaction of $O_3$ only takes place at the surface of the aerosol, which creates a shell of reactive products and limits further reactions (Berkemeier et al., 2021). Given the limited uptake of $O_3$ into oleic acid particles, similar

diffusion limitations could play an important role in the heterogeneous reactions of BBOA (Gerrebos et al., 2024). However, this has not definitively been shown in previous work on the heterogeneous reactions of BBOA, and warrants further investigation (Hems et al., 2021).

Here, we provide insight into the role of oxidative processing of BBOA with $O_3$ on the chemical composition of BBOA at a

wide range of RH. Experiments were performed with a series of BBOA sources from residential stoves (beech and spruce wood) as well as open burning of spruce branches. We have used two complementary online mass spectrometric techniques: the AMS and the extractive electrospray mass spectrometer (EESI-MS), to probe the changes in the bulk and molecular composition of primary BBOA particles, respectively. Specific molecular formula observed in the BBOA rapidly decayed when exposed to $O_3$ while a large portion of the BBOA constituents were relatively unreactive. We discuss how particle size

and relative humidity alter the extent of reactivity.

## 2 Experimental setup

The ozonolysis experiments of BBOA particles (Table 1) were conducted in the oxidative flow reactor (OFR; (Li et al., 2019)) and the 8 $m^3$ Teflon smog chamber, that are part of the atmospheric simulation chamber and burning platform located at the Paul Scherrer Institute in Villigen, Switzerland (Platt et al., 2013). In both setups, continuous measurements of the particles

were performed with a high-resolution time of flight aerosol mass spectrometer (HR-ToF-AMS, Aerodyne Inc.), an extractive electrospray ionization time of flight mass spectrometer (EESI-ToF-MS, Tofwerk), and a scanning mobility particle sizer (SMPS, TSI). Sketches of the complete setup are in the Supplement (Figs. S1-S2).



**Table 1: Overview over experiments. Note \*: in these experiments, the upper O₃ limit is an underestimation, as the O₃ monitor's upper measurement limit was restricted.  The true value is close to experiment n=8 or n=9; note entries with - : in these experiments, the O₃ concentration was at a background level or RH conditions were dry (~ 2% RH), respectively.**

| n | Date | Naming convention | Stove type | Fuel type | O₃ range [ppb hrs] | RH range [%] |
|---|---|---|---|---|---|---|
| **OFR experiments** | | | | | | |
| 1 | 02.03.2022 | spruce | Residential stove | spruce | 1.8 - 102.7 | - |
| 2 | 04.03.2022 | open | Open fire | pine | 1.8 - 108.4 | - |
| 3 | 09.03.2022 | beech | Residential stove | beech | 1.8 - 137.8 | - |
| 4 | 10.03.2022 | spruce2 | Residential stove | spruce | 2.7 - 144.4 | - |
| 5 | 11.03.2022 | open2 | Open fire | pine | 1.8 - 176.4 | - |
| 6 | 16.05.2022 | beech2 | Residential stove | beech | 0.4 - 88.9* | - |
| 7 | 17.05.2022 | spruceRH1 | Residential stove | spruce | 0.4 - 88.9* | 2 - 84 |
| 8 | 18.05.2022 | spruceRH2 | Residential stove | spruce | 0.9 - 107.1 | 2 - 91 |
| 9 | 20.05.2022 | open3 | Open fire | pine | 0.9 - 112 | 2 - 97 |
| **Smog chamber experiments** | | | | | | |
| 10 | 22.02.2022 | chamber_POA | Residential stove | spruce | - | - |
| 11 | 19.05.2022 | chamber1 | Residential stove | spruce | 15 – 2000 | - |

## 2.1 Ozonolysis procedure in the Oxidation Flow Reactor (OFR)

For each experiment, we generated fresh BBOA emissions from fires of either spruce, beech, or pine wood and injected the sample through heated stainless steel lines (150 °C) into a 1 m$^3$ insulated stainless steel holding chamber. Logs of spruce and beech wood were burned in a residential stove, the pine wood branches and needles were burned in an open stainless steel cylinder (65 cm diameter, 35 cm height) to mimic forest wild fire conditions. The holding chamber was filled with the desired emissions until the particle mass concentration reached 5 to 10 mg m$^{-3}$ (50 – 150 μg m$^{-3}$ in the sampling lines after dilution). While filling the holding chamber, a mix of emissions from flaming and smoldering conditions of the fires was sampled. The holding chamber served as a reservoir in which the primary BBOA and gaseous emissions were stored for the duration of the experiments at room temperature (20 °C) and from which we continuously sampled into the OFR. The emissions in the holding chamber were sampled at ~ 0.6 L min$^{-1}$ (detailed flow conditions in the supplement, Tables S1-S2) for the duration of the experiment with an equal make up flow of clean air (AADCO XX) to maintain a constant volume and pressure. The slow wall loss was corrected for in order to separate the decrease in particle mass concentration from changes due to O₃ exposure (see Section 2.3.3). In between experiments, the holding chamber was flushed with clean air to minimize carry-over of OA.



The holding chamber held both the primary BBOA and gas-phase VOCs emitted by the combustion sources, and to probe the reaction of BBOA with $O_3$ the VOCs were scrubbed with a charcoal denuder (Ionicon) at a flow rate of 0.6 L min$^{-1}$. In between experiments, the denuder was regenerated by heating to 200 °C while passing $N_2$ through the denuder to ensure a stable gas

removal efficiency during the experiment. Following denuding, the BBOA was exposed to $O_3$ in the OFR, a 50.8 cm long quartz-fused cylinder tube with a total volume of 16 L. More details on the OFR design are described elsewhere (Li et al., 2019). To achieve optimal flow conditions through the OFR of about 6 L min$^{-1}$, a flow of clean air into the OFR was added to the sample flow of ~ 0.6 L min$^{-1}$. An ozone generator provided the $O_3$ in the OFR, generating $O_3$ in situ by the photolysis of $O_2$, with a constant input flow of 0.1 L min$^{-1}$ clean air. By splitting the exiting flow from the ozone generator into an exhaust

and a line into the OFR, the $O_3$-containing flow in this line could be set between 0.01 and 0.1 L min$^{-1}$. Note that the total flow through the OFR was kept constant at 6 L min$^{-1}$ by adjusting the dilution flow of clean air. With the $O_3$ flow controlled between 0.01 and 0.1 L min$^{-1}$, the $O_3$ concentration inside the OFR varied between 0.02 and 4 ppm, as was continuously measured by an ozone gas monitor (Thermo 49C). The average residence time of particles in the OFR was 160 sec (16 L : 6 L min$^{-1}$), therefore this $O_3$ concentration range results in equivalent $O_3$ exposures between 1.4 and 176.3 ppb hrs. To control the exposure

time, we installed a second denuder directly after the OFR particle sampling outlet. In all experiments, the particles were sampled with an AMS, SMPS, and EESI-TOF.

In each experiment, we exposed the primary BBOA particles to 8 to 11 different $O_3$ concentration steps (Tables S1-S2). Three out of nine experiments included steps at different RHs increased compared to the background RH level of ~2 %. To increase

RH in the OFR, the clean air supply was split into a dry air line and a wet air line, which bubbled through a water bottle. At each step, the particles' chemical composition and size distribution after exposure to $O_3$ were monitored for at least 10 min after stable $O_3$ and RH conditions, respectively, were reached. Additional steps included sampling primary BBOA particles with no $O_3$ exposure to monitor the wall loss (Section 2.3.3).

**2.2 Smog chamber ozonolysis procedure**

A subset of experiments with longer $O_3$ exposures were accomplished by filling an atmospheric simulation chamber with the POA from a residential wood stove. The emissions, again, were passed through heated lines (180 °C) and prior to entry to the chamber they were passed through charcoal denuders in parallel to strip the gaseous aerosol species (VOCs). For all measurements, the non-methane VOCs were less than 10 ppbC from measurements with a Total HydroCarbon (THC) analyzer.

After injection into the chamber, $O_3$ was steadily ramped to 1 ppm and the experiment was allowed to proceed for 6+ hrs before the chamber was cleaned by flushing with zero air.



Sampling from the chamber for these measurements included an AMS, SMPS, gas monitors ($O_3$ – Thermo 49C, THC – Horiba APHA, CO, and $CO_2$ PICARRO).

## 2.3 Particle instruments

### 2.3.1 High-resolution time-of-flight aerosol mass spectrometer (HR-AMS)

The HR-AMS provides online quantitative measurements of the BBOA particle chemical composition (details in DeCarlo et al., 2006). Here, we operated the HR-AMS at a mass resolution of 2'000 to 4'000 m/$\Delta$m in V-mode using either a high-resolution ToF (H-ToF, lower resolution limit) or a long ToF (L-ToF, upper resolution limit). The time resolution for one measurement cycle was 1 min. For the conversion of ion counts into particle mass concentration of µg m$^{-3}$, we determined the ionization efficiency based on a calibration with 300 nm ammonium nitrate particles and simultaneous SMPS (model 3938, TSI) measurements. Data analysis of raw mass spectra was conducted in Igor Pro 8.0.4.2, using the SQUIRREL data analysis toolkit (version 1.63l) for unit mass resolution data up to 600 m/z and the PIKA module (version 1.23I) for high-resolution peak fitting in the range 12 to 170 m/z. To represent the composition at each $O_3$ step lasting a minimum of 10 min, we averaged the mass spectra of 11 to 13 measurement runs and report the standard error (standard deviation/(number of runs averaged)$^{1/2}$) at each m/z.

### 2.3.2 Extractive electrospray ionization time-of-flight mass spectrometer (EESI-ToF)

The EESI-ToF provides real-time, high-resolution measurement of OA on a near-molecular level (details in Lopez-Hilfiker et al., 2019). Incoming particles are sampled through a charcoal denuder to remove any gas-phase species and collide with electrospray droplets, which extracts water-soluble molecular constituents of the particles. Here, we used an aqueous solution of pure Milli-Q water doped with 100 ppm of sodium iodide (NaI). The solvent is evaporated rapidly from the droplets by passing the heated (~ 270 °C) capillary inlet to the mass spectrometer and analytes are ionized by forming adducts with Na$^+$. These ions are then sampled into an atmospheric pressure interface time-of-flight mass spectrometer for separation at a mass resolution of ~ 10'500 m/$\Delta$m (APi-ToF, Tofwerk, Thun, Switzerland). Mass spectra were recorded in the positive mode with a time resolution of 1 sec. For data processing, the mass spectra were pre-averaged over 10 sec. After averaging, the HR data analysis was performed using the Tofware software version 3.2.3 for Igor Pro (8.0.4.2), fitting ions up to 420 m/z.

The continuous measurement alternated between sampling the incoming aerosol for 6 min and recording the ambient background signal for 90 sec by sampling the incoming flow through a high volume HEPA filter. To retrieve the background-corrected average intensity of each ion with a time resolution of 7 to 8 min, the average intensity during a filter period was subtracted from the average intensity during the preceding signal period. For the composition representing one $O_3$ exposure condition, we averaged the ion intensities over 2 to 4 signal periods and report the error propagated from the standard errors



of all signals periods in that $O_3$ period. Species were considered for the analysis if their intensity was > 10 cps in all relevant signal periods.

### 2.3.3 Wall loss correction

There are two loss mechanisms (dilution and wall loss) for particles in holding chamber. Additionally, gaseous species are absorbed to the walls, driving their partitioning out of the particulate phase at a rate depending on their volatility (Fig. S4). To separate these decreases in particle concentration from potential mass loss due to $O_3$ exposure, we applied a wall loss correction to all measured particle concentrations (HR-AMS, SMPS) and individual ion intensities (EESI-ToF) for the OFR experiments (Eq. 1, Fig. S4).

$$var_{corr}(t) = var_{measured}(t) + var_0(t) - var_{fit}(t) \quad\quad (1)$$

In Eq. 1, subscript *corr* refers to the final corrected value for any variable at time point *t*, *measured* is the value as measured on the instrument, *0* is the average concentration or intensity during the first stable period of POA and *fit* refers to the value retrieved from a linear fit through periods of POA. This linear fit was adequate to capture both the particle loss dynamics and repartitioning losses for the duration of the ozonolysis experiments. For experiments where a higher flow demand was required

to supply additional particle instruments, we applied an exponential fit instead of a linear fit (experiments beech2, spruceRH2, Fig. S3).

## 3 Results

### 3.1 Primary BBOA particle composition

#### 3.1.1 HR-AMS data

The HR-AMS mass spectra of primary BBOA particles exhibit consistent dominating features across burning and fuel types (representative example from experiment spruce2 in Fig. 1a and Fig. S6) and match qualitatively with HR-AMS measurements of primary BBOA from previous studies (Alfarra et al., 2007; Fortenberry et al., 2018; He et al., 2010; Zhang et al., 2023). The chemical class of CH fragments, characterized by ion series of $C_nH_{2n-1}$, makes up the largest fraction of the total particle mass concentration, accounting for $45.9 \pm 0.1$ to $58.2 \pm 0.1$ % across all experiments. Other prominent peaks include m/z 29,

43, 44, and 60 resulting from the contribution of oxygen-containing ions $CHO^+$, $C_2H_3O^+$, $CO_2^+$, and $C_2H_4O^+$, that are typical fragments of BBOA compounds (Alfarra et al., 2007; Fine et al., 2002; Lin et al., 2016; Ortega et al., 2013; Zhang et al., 2023). We also consistently found a clear signal up to 2.7 % for the BBOA marker species at m/z 60, $C_2H_4O_2^+$, an ion resulting from the pyrolysis of anhydrous sugars like levoglucosan (Aiken et al., 2009; Cubison et al., 2011; Lee et al., 2010; Simoneit et al., 1999). Though we do observe variations in relative fractions of $f_{60}$ up to 0.8% between experiments (Figure 2c-e), these are



likely based on variations in the fuel material and burning conditions and are within the variability for complex burning experiments.

### 3.1.2 EESI-ToF data

In the EESI-ToF mass spectra of primary BBOA particles, we identified a small number of distinct, outstanding peaks recurring across fuel and burning types in the fitted mass range up to 420 m/z (example from experiment spruce2 in Fig. 1b). As a
particular characteristic for BBOA, the most intense peak was consistently measured for $C_6H_{10}O_5$ with relative intensities up to 30.2 %. This molecular formula corresponds to the marker species levoglucosan and its isomers (162.0523 m/z), which has been frequently matched with biomass burning sources before (Kumar et al., 2022; Qi et al., 2019; Stefenelli et al., 2019; Zhang et al., 2023). In all but one experiment (open2), second-highest contributions are found for $C_8H_{12}O_6$, which can likely be attributed to derivatives of syringol, a monomeric subunit of the biomass polymer lignin (Yee et al., 2013). Lastly, groups
of $C_{20}$ species around 302 m/z and 318 m/z are highlights of the EESI-ToF POA mass spectra. Possible sources of these $C_{20}$ species are discussed in more detail in Section 3.3.1. Also note, that although polyaromatic hydrocarbons (PAHs) are another unique group of molecules typically present in POA (Bozzetti et al., 2017), the current configuration of the EESI does not measure these molecules because either their solubility or binding dynamics with $Na^+$ preclude their detection.

This group of dominating ions ($C_6H_{10}O_5$, $C_8H_{12}O_6$, and $C_{20}$'s) remained largely the same despite overall variability in the
intensities of individual ions between each experiment's POA spectra (Fig. S7). This POA variability is likely the result of inherent variations between experiments, driven by diversity in the fuel material (age of the logs, amount of resin, etc.) and specifics of the burning conditions such as the exact duration of flaming and smoldering stages. As extensive fragmentation occurs with the HR-AMS, this POA variability is only reliably detected with the EESI-ToF, highlighting the complementary use of both instruments (Zhang et al., 2023).

## 3.2 Bulk composition changes of primary BBOA upon $O_3$ exposure

### 3.2.1 HR-AMS data

Without $O_3$ exposure, the oxygen-to-carbon (O/C) ratios of primary BBOA particles range from 0.30 to 0.43, increasing from pine to spruce to beech (Fig. 2a). This order is reproducible across all experiments performed and thus the O/C ratio is useful as a characteristic bulk feature for each fuel type. The highest O/C ratios for beech wood POA also coincide with highest mass
fractions of oxygenated chemical classes $C_xH_yO_1$ and $C_xH_yO_{2+}$ of 40.6 ± 0.1 % for this fuel, while the average of all experiments was at 33.5 ± 0.1 %. The range of O/C ratios measured here is comparable to previous measurements of primary BBOA (e.g. Aiken et al., 2009; Xu et al., 2020).

Upon exposure to $O_3$, the O/C ratio increases for all conducted experiments, which is indicative of oxidative aging in the particle phase (Fig. 2a). The increases in O/C ratio from POA conditions to the highest $O_3$ exposure of each experiment are





similar throughout, even though the experiments differed in maximum $O_3$ exposure tested. The range of increase is between 0.03 (experiment open2) and 0.06 (experiment spruce2), which is equivalent to an increase of 15 % in the particles' oxygen content. For all experiments, the increase in O/C is rapid, occurring at $O_3$ exposures below 50 ppb hrs (equivalent to 1 hour of a typical summer day in the atmosphere), which is well below the maximum $O_3$ exposure of any experiment (176 ppb hrs). In experiments performed in the smog chamber, we tested $O_3$ exposures up to ten times the OFR range, i.e. up to 2000 ppb

hrs. Similar to the OFR measurements, there was a fast increase in the O/C in the first minutes of exposure, and an upper limit was reached after ~ 3 hrs of $O_3$ exposure. In comparison, during a background POA experiment with no $O_3$, the O/C increased slowly over 2-3 hours to +10 % of the initial value (chamber_POA, Table 1, Fig. S8), presumably from evaporation of semi-volatile species. After ten-fold increase of $O_3$ exposures in the smog chamber, the O/C ratio increase remains similar as in the OFR experiments (Fig. 2b). After a steep increase up to ~ 50 ppb hrs, the slope gets shallow, which could either mean that the

reactive species present in the POA are fully consumed after this period of time, or that there could be a diffusion limitation governing the maximum degree of oxidation in the ozonolysis experiments at dry conditions, slowing down the oxidation process. These possibilities will be explored below in the atmospheric implications section.

Fig. 2c-e also illustrates an overall increase in the oxidative state of the particles upon $O_3$ exposure based on an increase in $f_{44}$. $f_{44}$ is the mass fraction of m/z 44 which is mostly $CO_2^+$, i.e. the end of an oxidative chain. Across fuel types and repeated

experiments, $f_{44}$ increases by a maximum of 1.2 % on the scale of total mass, comparing the fraction at each experiment's maximum $O_3$ exposure to the fraction in fresh POA. In contrast, $f_{60}$ ($C_2H_4O_2^+$) remains largely stable throughout all $O_3$ exposures. As this ion is largely derived from levoglucosan, this stability implies that levoglucosan is not reactive towards $O_3$, and is consistent with its lack of double bonds on the carbon chain. The increase in $f_{44}$ and O:C must therefore result from the oxidation of other chemical moieties within the primary BBOA particles that contain double bonds, which are reactive with

$O_3$.

Based on the evolution of O/C ratio and $f_{44}$ vs. $f_{60}$, the effect of $O_3$ on the bulk composition of primary BBOA particles could be clearly and reproducibly observed. Under dry conditions, the degree of oxidation increases in the particles upon exposure to $O_3$, although this change is minor. These findings are consistent with previous measurements performed in Browne et al. (2017), where limited changes occur under relatively dry conditions (~2 % RH from our results vs. 30% RH).

**3.2.2 EESI-ToF data**

To identify the groups of species on a molecular level that drive the observed bulk reactivity towards $O_3$ (Section 3.2.1), we binned the species detected by the EESI-ToF by number of C and O atoms. The intensity of each bin was scaled relative to the total intensity in Fig. 3 and Fig. S9, and this classification shows that most species contain 5 to 20 C atoms and 0 to 7 O atoms. The dominant peaks in the mass spectra (Fig. 1b) stand out at 6 and 8 C atoms with the largest contributions from the single

species $C_6H_{10}O_5$ and $C_8H_{12}O_6$, respectively.





Contrasting POA conditions (Fig. 3a-c) to the changes in relative intensity at the maximum $O_3$ exposure (102-176 ppb hrs) of each experiment (Fig. 3d-f) reveals that the same groups of species for all fuel types are strongly reactive towards $O_3$. Most prominently, species containing 18 and 20 C atoms with 1 to 3 O atoms decay significantly upon exposure to $O_3$. In particular, the intensity of $C_{18}H_{xx}O_2$ species decreased by $44.2 \pm 0.3$, $83.5 \pm 0.3$ and $66.2 \pm 0.1$ % compared to their initial intensity in the

experiments beech, spruce, and open2, respectively. Similarly, $C_{20}H_{xx}O_2$ species decayed by $29.2 \pm 0.1$ (beech), $58.4 \pm 0.1$ (spruce) and $36.9 \pm 0.1$% (open2). As a third important group, $C_{16}$ species are among the highest decaying species, for example in the experiments spruce and open3 (Fig. S9). Most species that are not $C_{16}$, $C_{18}$, or $C_{20}$ contribute each less than 1% of relative change to the total POA intensity at maximum $O_3$ exposure. The higher variation of species like levoglucosan, $C_6H_{10}O_5$, are considered as within errors related to wall loss correction.

When comparing the experiments further, three observations are especially evident:

(1) Even though the experiment with the highest $O_3$ exposure overall was open2 (176 ppb hrs), the strongest decrease of the reactive species groups was observed in experiment spruce, where maximum $O_3$ was at 102 ppb hrs. This observations points at parameters other than the $O_3$ exposure (Section 3) determining the progression of the ozonolysis; and this limitation starts at an $O_3$ exposure threshold below the maximum exposures tested in the

experiments here.

(2) The beech experiment exhibited the lowest change upon $O_3$ exposure, which matches to the POA composition being more oxygenated than spruce or pine, and therefore the compounds therein being the least reactive. The higher oxygenated state at POA conditions shows in the order of O/C ratios at POA conditions (Fig. 2a) and the above average abundance of the chemical classes $C_xH_yO_1$ and $C_xH_yO_{2+}$. For example, species containing 4 or more O atoms

(excluding $C_6H_{10}O_5$) make up 49.1 % of the total POA intensity in beech, compared to 24.5 % in experiment spruce and 36.8 % in open2 (Fig. 3).

(3) A significant number of product species was only observed in experiment open2 (Fig. 3c). One possible reason for the overall lack of detection could be that many ozonolysis products are volatile, evaporate from the particle and therefore evade detection.

Overall, the EESI-ToF bulk analysis of the dry OFR experiments reveals that groups of low oxygenated species containing 16, 18, and 20 C atoms drive the $O_3$ reactivity within primary BBOA particles. These identified groups of species consistently decay upon $O_3$ exposure and are present across all tested fuel types, albeit with varying relative abundance (higher abundance in pine and spruce combustion emissions). The exclusivity of $O_3$ reactivity to a small subset within the total POA composition matches well with the small degree of overall changes that were observed in the bulk HR-AMS characteristics (Section 3.2.1).





### 3.3 Reactivity of primary BBOA during ozonolysis on a molecular level

#### 3.3.1 Key decaying species

Within the groups of $C_{20}H_{xx}O_2$ and $C_{18}H_{xx}O_2$, the most important contributors in all dry OFR experiments to the reactivity towards $O_3$ are the species $C_{20}H_{28}O_2$, $C_{20}H_{30}O_2$, $C_{18}H_{32}O_2$, and $C_{18}H_{34}O_2$. These species have the highest absolute signal intensity and the highest intensity decrease at maximum $O_3$ exposure within their groups. Analysing their individual decay as function of $O_3$ exposure further mirrors the trend in the O/C ratio observations (Fig. 2a). The species' intensity drops steeply up to 50 ppb hrs followed by a decrease with shallow negative slope at higher $O_3$ exposures (Fig. 4). At maximum $O_3$ exposures and across experiments, fractions down to 29.0%, 12.6% and 12.2% remain for $C_{20}H_{30}O_2$, $C_{18}H_{32}O_2$, and $C_{18}H_{34}O_2$, respectively. The remaining fraction of $C_{20}H_{28}O_2$ at highest $O_3$ exposure is on average ~ 20 % higher than for $C_{20}H_{30}O_2$.

The molecular formulas $C_{20}H_{30}O_2$, $C_{18}H_{32}O_2$, and $C_{18}H_{34}O_2$ could be connected to abietic acid, linoleic acid, and oleic acid, respectively (Fig. S10). Abietic acid-like structures are known components of resins and needles in conifers (Iinuma et al., 2007; Mofikoya et al., 2020; Schauer et al., 2001; Liang et al., 2021, 2022), which is consistent with their higher abundance in pine and spruce combustion emissions. Linoleic and oleic acid have also been detected in emissions from pine wood burnings (Nolte et al., 2001), but have been measured in other emissions, e.g. from cooking, too (Robinson et al., 2006). All three compounds are unsaturated, explaining their observed reactivity with $O_3$. Oleic acid in particular has often been applied as model compound for ozonolysis (Berkemeier et al., 2021). We also detected, presumably, de-hydroabietic acid ($C_{20}H_{28}O_2$) in similar abundance to $C_{20}H_{30}O_2$, which showed some reactivity towards $O_3$, but smaller than the other molecules noted in this section. One explanation could be the presence of aromatic systems in $C_{20}H_{28}O_2$ with de-located electrons, where $O_3$ is less prone to attack the molecule compared to the double bonds as in $C_{20}H_{30}O_2$.

Based on the experiments conducted at dry conditions, we outlined the reactivity of BBOA particles from spruce, pine, and beech fires towards $O_3$ and identified the drivers of this reactivity on a molecular level. In all analysed parameters, most of the oxidative changes by $O_3$ was reached at exposures around 50 ppb hrs in the OFR for both the bulk changes (Fig. 2) and molecular changes (Fig. 3 and 4), with slow progression at higher $O_3$ exposures. We hypothesize that the diminished effect at $O_3$ exposures exceeding that threshold could be due to either the diffusion of $O_3$ through the BBOA particle starting to limit the further progression of the ozonolysis on the timescale of OFR experiments or due to different isomers present in the POA with some isomers reactive to $O_3$ and others that are unreactive. The diffusion limitation of molecules in BBOA could either be related to the diffusivity of primary BBOA itself, or via reaction products of $O_3$ creating a shell with low diffusivity, which has been observed in pure oleic acid aerosol (Berkemeier et al., 2021). Sections 3.3.2 and 3.3.3 discuss the effect of relative humidity and relevance of particle size supporting the hypothesis of the diffusion limitation for the progression of the ozonolysis.





### 3.3.2 The effect of relative humidity on the reactivity of primary BBOA

To test the hypothesis that diffusion of $O_3$ is limiting the progression of the ozonolysis, we performed a subset of experiments comparing dry conditions to 50% and > 80% relative humidity at distinct exposures of $O_3$. Increasing RH should enhance the diffusion with the POA particles, as they become less viscous due to the uptake of water (e.g. Asa-Awuku et al., 2009; Reid et al., 2018). Diffusion rates of small molecules (e.g. $O_3$) through a particle increases from ~ $10^{-10}$ cm$^2$ s$^{-1}$ in glasses/solids to ~ $10^{-5}$ cm$^2$ s$^{-1}$ in liquid particles (Shiraiwa et al., 2011). If diffusion limits the extent of ozonolysis, increasing the RH will mitigate that limit and enable the ozonolysis to advance further.

Figure 5 compares the bulk O/C ratio change for each type of POA for different $O_3$ exposures and RHs. Indeed, the O/C increases more during periods with increased RH than at dry conditions, and the additional increase was largest at 90 ppb hrs $O_3$ exposure. Comparing the difference in O/C ratio change between low and high $O_3$ exposure, the difference with changing RH was negligible (experiment spruce RH1) in the low $O_3$ exposure range, while at high exposures there is a greater change in the ΔO/C at high humidity conditions. At high exposures of $O_3$, potentially all POA molecules that are reactive to $O_3$ and are at the surface of the particle do react quickly with it. Then the diffusion limitation becomes relevant; for the ozonolysis of the POA particle to progress further, $O_3$ needs to diffuse further into the particle to reach more POA reactive molecules or these molecules need to diffuse to the surface to react with $O_3$. The observation is consistent with diffusivity (for either $O_3$ or reactive organic molecules) being increased at elevated RH.

Figure 6a-c shows the EESI-ToF data for $C_{20}H_{30}O_2$, $C_{18}H_{32}O_2$, and $C_{18}H_{34}O_2$ for experiment spruceRH1. For these species that react away with $O_3$ in the particles (Section 3.3.1), the increase in loss at elevated RH is especially evident and illustrates that ozonolysis proceeds further under wet conditions. For example, at 90 ppb hrs $O_3$ exposure, the remaining fraction of $C_{18}H_{32}O_2$ at 50% and 80% RH is reduced to 11.1% and 16.9% compared to POA conditions, whereas 80% remain at the same $O_3$ exposure under dry conditions. A similar pattern is observed for $C_{20}H_{30}O_2$ and $C_{18}H_{34}O_2$. The changes in the extent of ozonolysis of BBOA with changing RH observed by the EESI-ToF reinforces the observations of the O/C ratio changes in Figure 5 supporting the diffusion limitation hypothesis.

In addition to increasing the diffusion of $O_3$ into the BBOA particle, also species within the particle could diffuse to the surface faster and evaporate out of the particle to achieve equilibrium between gas and condensed phase at wet conditions. As a result, a large fraction of volatile, lower-molecular weight species containing 6-14 C atoms are lost. In experiment spruceRH1, the overall EESI intensity at 50% and 80% RH decreased by 39.7% and 29.8%, respectively, compared to dry conditions at 90 ppb hrs $O_3$ exposure (Fig. S12). As the current background reference at 4 ppb hrs $O_3$ exposure also shows a similar decay at wet conditions, the observed loss is likely not connected to the ozonolysis. Additionally, this effect is illustrated by the species $C_8H_{12}O_5$, $C_9H_{12}O_3$ and $C_{10}H_{14}O_3$ from experiment spruceRH1 as the largest contributors to intensity of the lower-molecular weight species (Fig. 6d-f). Their intensity decreases by up to 76% compared to POA conditions when conditions are wet. The difference in the remaining fractions for all three species is negligible between 4 and 90 ppb hrs $O_3$ exposure. Therefore, the





loss is not the result of the ozonolysis, but rather enhanced evaporation. Further, enhanced evaporation under humid conditions additionally supports the hypothesis that molecular diffusion is limited within primary BBOA under dry conditions, but that
at elevated RH these limitations are eased.

### 3.3.3 Particle size dependence

Investigating the particle size dependence of the POA ozonolysis was possible as the SMPS measured continuously and as the average POA particle size increases with increasing residence time in the holding chamber. This increase in average size is predominantly due to wall loss of small particles and coagulation of smaller particles to form larger particles.
As diffusion time scales increase with particle size, we expect the heterogeneous ozonolysis of smaller particles to progress further than of larger particles for the same reaction duration. That is, if there is indeed a diffusion limitation.

Figure 7 supports this hypothesis based on changes in O/C ratio measured by the AMS for the experiments open2, open3 and spruce2. In all three experiments, the O/C ratio increased more in the smaller particles than in the larger particles upon $O_3$ exposure. Comparing the experiments, the difference in O/C ratio change further increased with the difference in particle size.
In experiment open3, the size difference was smallest (5 nm) and the difference in O/C change remained within the range of uncertainty. By contrast, in experiment spruce2 the diameter increased by 32%, and here, the increase in O/C ratio compared to POA conditions was double (0.06) for the smaller particles than the larger particles (0.03).

Notably, the $O_3$ exposures of the experimental steps compared in Fig. 7 were not constant, ranging maximum 30 ppb hrs (details see caption of Fig. 7). Therefore the differences in O/C change are driven both by a changing $O_3$ exposure and by a
changing particle size. However, all exposures were above 50 ppb hrs (minimum 91 ppb hrs), the threshold previously demonstrated as the limit for $O_3$ exposure dependence for the ozonolysis under dry conditions in the OFR. Based on this result, we argue that the driving factor for the differences in O/C ratio change in Fig. 7 is indeed the differences in particle size.

Overall, these trends support the role of a diffusion limitation in the dry, heterogeneous ozonolysis of POA, with smaller particles exhibiting greater increases in O/C ratio compared to larger particles under the same reaction conditions.

## 375    4 Atmospheric implications

The results presented herein provide comprehensive chemical measurements of the heterogeneous reactions of $O_3$ with authentic primary biomass burning organic aerosol generated from residential stoves or open burning processes. Overall, these results show that the ozonolysis proceeds only to a limited extent under dry conditions, consistent with previous results. In all cases the $O_3$ uptake into primary BBOA is limited to the surface under dry conditions at 20 °C, but is eased when exposed to
elevated RH. The change in O/C ratio because of heterogeneous uptake is always very minor. This means one cause of the limitation is the lack of reactive species toward $O_3$. The other point driving the limitation under dry conditions is the molecular diffusion limitation. Both the diffusing of $O_3$ into the particle and diffusing of species towards the surface are limited within



BBOA under dry conditions. In the atmosphere, there are limited places where strong BBOA emissions occur with low relative humidity at ~20 °C, but BBOA emitted from wildfires can frequently reach the troposphere and on rare occasion the
stratosphere (Wilkins et al., 2020). When strong convection is coupled with the emissions from wildfires the ambient temperatures will decrease below 10 °C, creating conditions where BBOA is mostly likely a glassy solid with viscosities above $10^{12}$ Pa sec (Arangio et al., 2015). In these atmospheric conditions, the uptake of $O_3$ will be even more limited to the surface because of the higher viscosities and lower diffusivity.

Previous work has shown that the reactions of oleic acid with $O_3$ results in a high viscosity shell on the surface of particles,
which then limits the diffusion of molecules reaching the surface (Berkemeier et al., 2021). Because the oxidation state changes observed here are small, and these species make up a small fraction of the measured aerosol, we hypothesize that there are insufficient amount of oxidation products formed capable of creating the low diffusivity layer that limits molecules diffusion to the surface. Future work will focus on investigating the dynamics across a wide range of $O_3$ exposures to elucidate molecular changes under different relative humidities (e.g. 0, 50, 90%). Within this work, the changes with different $O_3$ exposures were
only investigated in detail under dry conditions, but specific experiments to target changes with RHs were performed to determine the effect of RH on the extent of the reactivity of primary BBOA. Having detailed chemical changes at a series of RHs will make it possible to model the diffusivity of $O_3$ and its changes with RH, similar to the modelling performed within Berkemeier et al., 2021 which will provide constraints on the diffusivity within primary BBOA. Within this work, we focus only on qualitative terms regarding diffusion limitations because not enough data is available to provide insight at different
RHs. Though, this work provides important insight into the diffusion limitations present within authentic primary BBOA and the chemical changes associated with heterogeneous reactions of $O_3$. This work answers important questions about the phase state of primary BBOA and future work should investigate quantitative details regarding molecular diffusion rates.

**Data availability**

Data from the figures may be found at DOI: 10.5281/zenodo.14721551

**Supplement**

**Author contributions**

Flow tube experiments were performed by SB and JZ with support from DMB, chamber measurements were performed by SB, RC, and DMB, conceptualization of the measurements were achieved by DMB, KL, ASHP, and IEH, the data was analysed
by SB, the manuscript was written by SB and DMB, all authors read and agree with the submission of the manuscript.



**Competing interests**

The authors declare that they have no conflict of interest.

**Disclaimer**

Publisher's note: Copernicus Publications remains neutral with regard to jurisdictional claims in published maps and
institutional affiliations.

**Acknowledgement**s

We are grateful for our technician Pascal Schneider who built the holding chamber and the associated equipment with the flow
tube.

**Financial support**

This work was supported by the Swiss National Science Foundation (SNSF) grant MOLORG (200020_188624), and SNSF
joint research project (189883), the ATMO-ACCESS Integrating Activity under grant agreement no. 101008004, and the
facility is part of ACTRIS ERIC and receives funding from the Swiss State Secretariat for Education, Research and Innovation
(SERI).

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





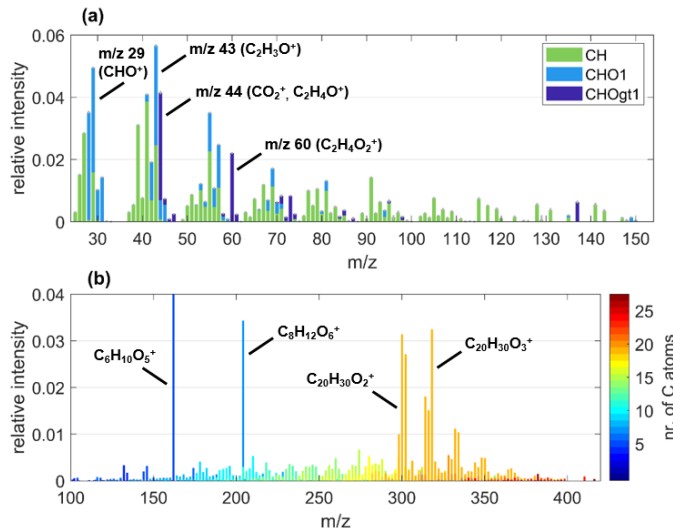

**Figure 1: Exemplary (a) AMS and (b) EESI mass spectrum at POA conditions of BBOA particles from experiment spruce2 highlighting important ions. The y-scale intensity is in (a) relative to the average POA mass concentration summed over each nominal mass up to m/z 170 and in (b) relative to the total average intensity at POA conditions (b). Note that in (b), the most intense fraction of $C_6H_{10}O_5$ would expand the y-scale by factor 3 (relative intensity of 0.17), so this ion was not considered for setting the scale.**

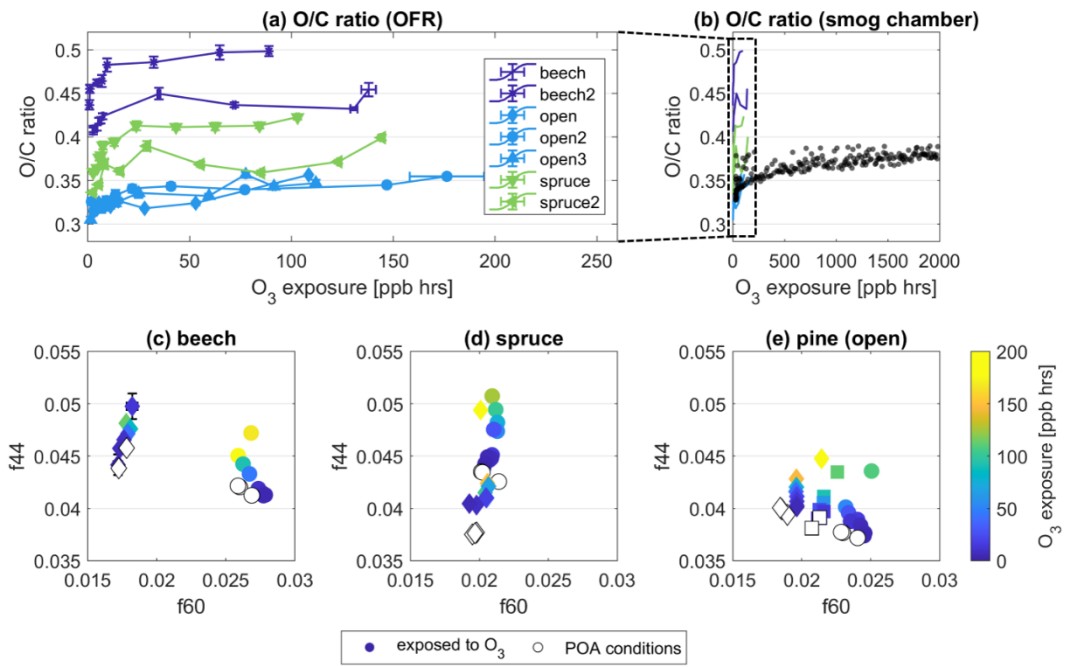

**Figure 2: Evolution of O/C ratio measured by the AMS in primary BBOA particles as function of $O_3$ exposure in (a) OFR experiments, (b) a smog chamber experiment with spruce wood. (c), (d), (e) evolutions of fraction m/z 44 (f44) vs m/z 60 (f60) from**



OFR experiments measured by the AMS and color-coded by O₃ exposure. Each subplot (c), (d), (e) combines data from all experiments for each fuel type given in the subtitle, with different markers for each experiment.

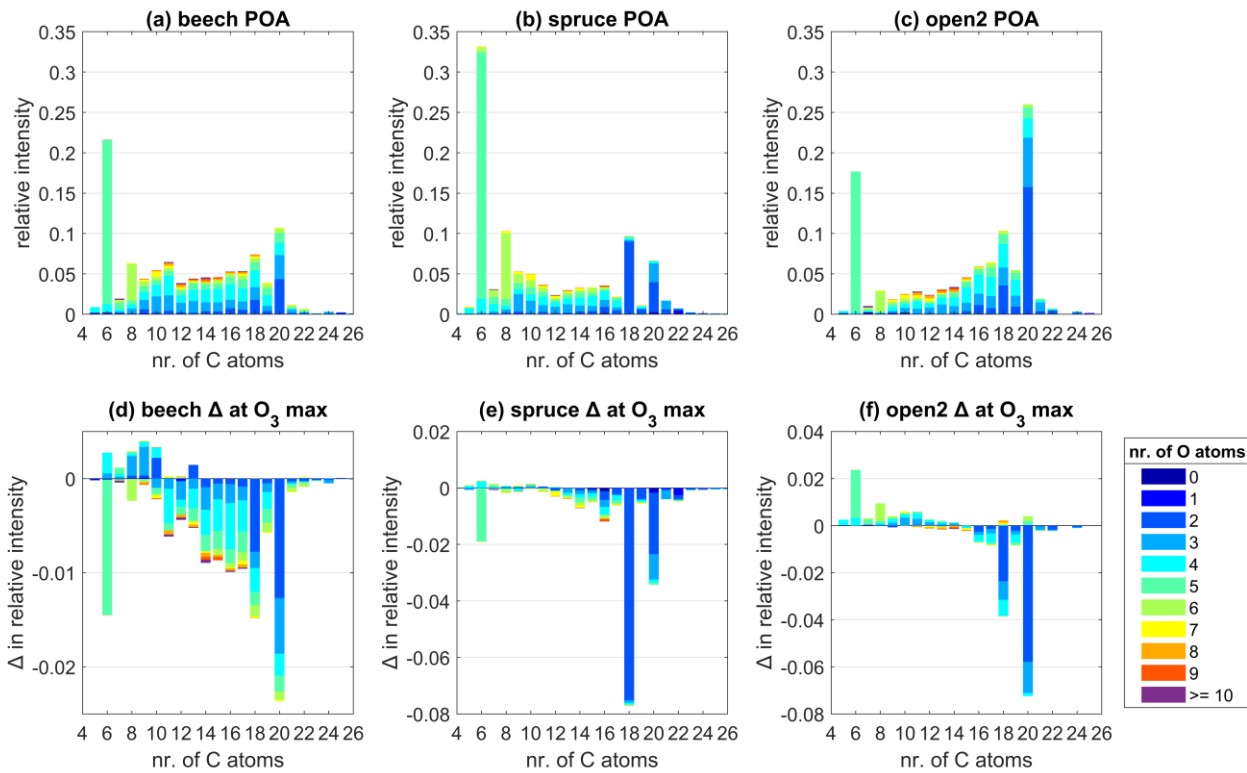


**Figure 3: Average POA composition measured by the EESI for the experiments (a) beech, (b) spruce, and (c) open2, classified by nr. of C atoms, color-coded by nr. of O atoms and scaled as relative intensity compared to total intensity measured at POA conditions. (d) – (f) Change in relative intensity at highest O₃ exposure of each experiment compared to POA conditions. The highest O₃ exposure varied between 137 ppb hrs ((d) beech), 102 ppb hrs ((e) spruce), and 176 ppb hrs ((f) open2).**





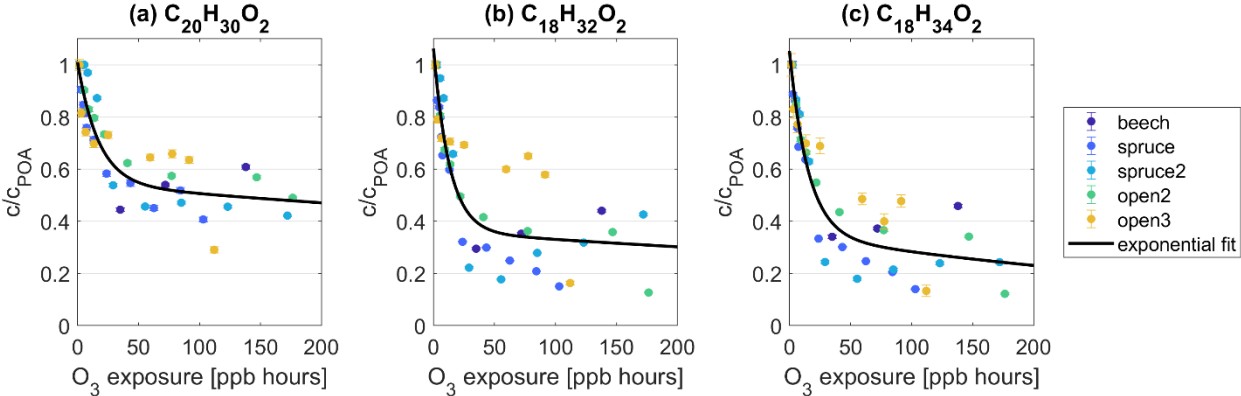

**Figure 4: Intensity c of species (a) $C_{20}H_{30}O_2$, (b) $C_{18}H_{32}O_2$, (c) $C_{18}H_{34}O_2$ as fraction of their intensity at POA conditions $c_{POA}$ and as function of $O_3$ exposure. The data points from all experiments were combined and are fitted with an exponential function of $c/c_{POA} = a * \exp(b * p_{O3} * t) + c * \exp(d * p_{O3} * t)$. The fit coefficients for $C_{20}H_{30}O_2$ ($C_{18}H_{32}O_2$, $C_{18}H_{34}O_2$) are a = 0.46 (0.70, 0.70), b = - 56 (- 76, - 65) * $10^3$ (ppb hrs)$^{-1}$, c = 0.54 (0.36, 0.35), and d = - 0.7 (- 0.9, -2) * $10^3$ (ppb hrs)$^{-1}$. $p_{O3}$ is the $O_3$ concentration in ppb, t = 160 s is the reaction time in the OFR.**

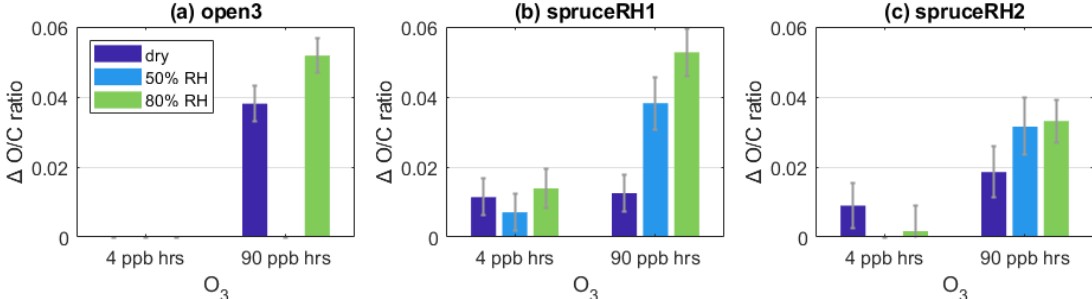

**Figure 5: Change in O/C ratio compared to POA conditions at low (0.1 ppm / 4 ppb hrs) and high (2 ppm / 90 ppb hrs) $O_3$ exposure at varying RH for the experiments (a) open3, (b) spruceRH1, and (c) spruceRH2. The exact $O_3$ and RH conditions are given in Tables S1-S2.**






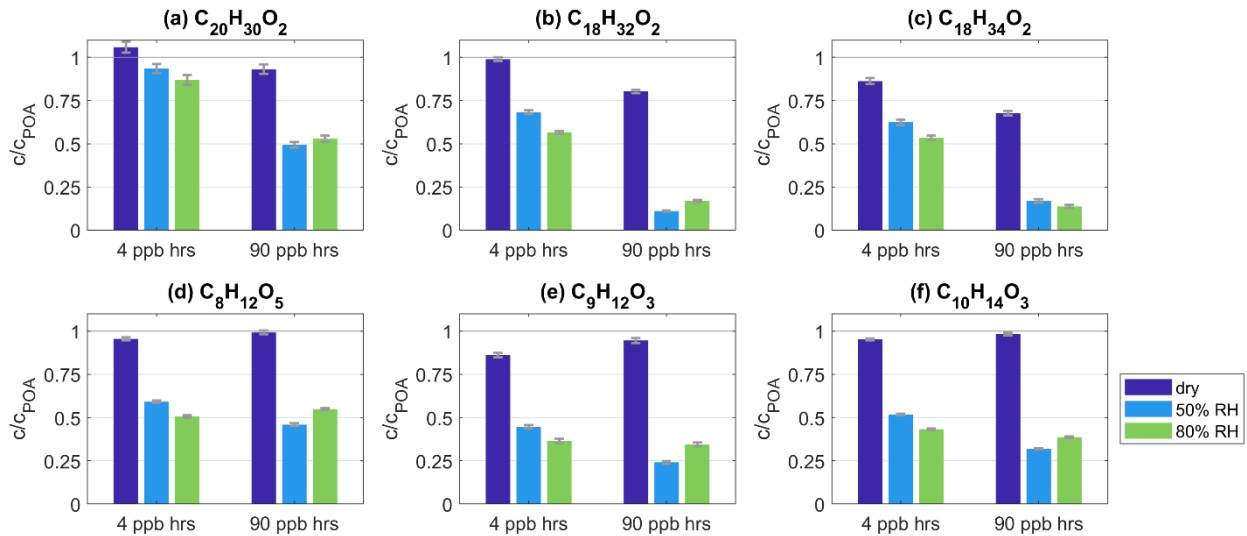

**Figure 6: (a)-(f) Intensity c as fraction of intensity at POA conditions $c_{POA}$ and as function of varying $O_3$ exposure and RH level. Each subplot shows the data for one species as given in the subtitle from experiment spruceRH1. The exact $O_3$ and RH conditions are given in Tables S1-S2.**

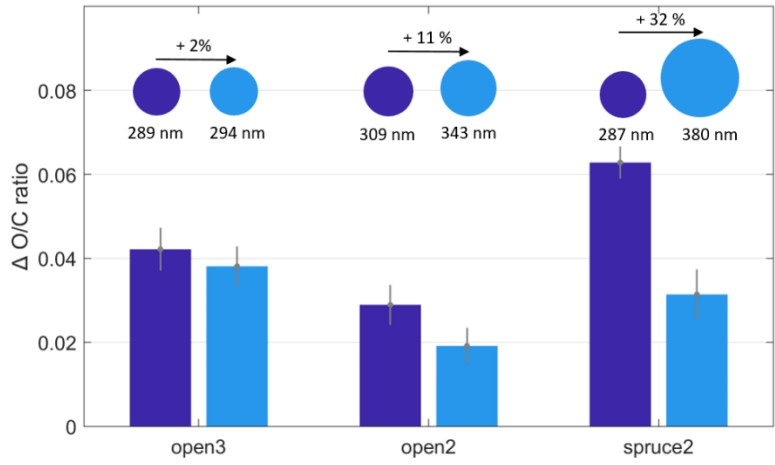


**Figure 7: Change in O/C ratio compared to POA conditions at varying particle size and $O_3$ exposure. The $O_3$ exposure varied between 91 or 112 ppb hrs (open3), 146 or 176 ppb hrs (open2) and 144 or 173 ppb hrs (spruce2). The particle size is given as the average particle geometric mean diameter.**
