# Peer review of "Ozonolysis of primary biomass burning organic aerosol particles: Insights into reactivity and phase state"

_EGUsphere, 2025_

## Referee Comment (RC1)

**General comments:**

Bogler et al. present an insightful study on the ozonolysis of biomass burning organic aerosols (BBOA) using flow reaction systems coupled with advanced mass spectrometry analysis. The manuscript thoroughly examines the chemical evolution of BBOA under varying relative humidity and ozone exposure conditions, offering a well-supported mechanistic discussion. This research addresses an important topic related to the atmospheric fate of wildfire and prescribed burning emissions, which play a crucial role in the atmosphere-climate system. The manuscript is well-structured, clearly written, and includes a detailed discussion of the results. Therefore, I believe it is well-suited for publication in *Atmospheric Chemistry and Physics* following minor revisions. Below, I provide a few specific comments that the authors may find helpful in further strengthening their study.

**Specific comments:**

Line 50: It may be worth noting that sunlight can also drive triplet-state chemistry and secondary oxidant formation inside BBOA particles, contributing to oxidative processes (Liang et al. 2024). Additionally, recent fieldwork by Vasilakopoulou et al. (2023) reported the rapid oxidation of biomass burning plumes, which could provide further context.

Line 75: Given the discussion on $O_3$ diffusion limitations in BBOA, it would be beneficial to elaborate on how liquid-liquid phase separation (LLPS) can lead to the formation of a highly viscous shell, which may further restrict oxidant penetration (Gerrebos et al. 2024; Gregson et al. 2023).

Line 80: It would be helpful to provide additional context on the prevalence of these fuel types in ambient environments, particularly in wildfire-prone regions, to strengthen the relevance of the study.

Line 100: Since combustion conditions significantly influence BBOA composition, mentioning the typical burning temperatures in real wildfire and prescribed burn scenarios would enhance the discussion.

Line 108: Is AADCO XX the brand of the gas generator?

Line 160: A discussion on how the low solubility of certain BBOA components might affect the performance and sensitivity of the EESI-ToF analysis would be valuable.

Line 240: It would be interesting to see whether the DBE plot captures the depletion of high DBE compounds upon oxidation, which could further support the loss of reactive species.

Line 248: Knopf, Forrester, and Slade (2011) reported ozone uptake by LEVO particles, though the detailed chemical mechanism remains unclear. A brief mention of this could provide additional context.

Line 282: The statement on volatility changes upon oxidation could be further elaborated, as increased volatility is often expected but depends on the specific reaction pathways involved.

Line 299: Were any aromatic compounds with vinyl side chains, such as coniferyl alcohol, detected? These species are commonly found in BBOA and are known to undergo ozonolysis (Fleming et al. 2020; Huang et al. 2021).

Line 322: Gregson et al. (2023) suggested that the BBOA can undergo LLPS to form a highly-viscous shell.

Line 344: Did the author refer to 'mass accommodation' and 'bulk phase diffusion to reach mass equilibrium'?

Line 391: The discussion on interfacial processes could be expanded, as the interfacial layer (~0.15 nm thick) in a 100 nm particle would only account for approximately 1% of the total particle volume. A quantitative perspective on this aspect would be helpful.

Line 400: The discussion on phase state as a key factor in multiphase oxidation is highly relevant. However, it would be useful to briefly comment on the oxidation of BBOA by other oxidants, such as OH radicals. Since BBOA has limited reactivity toward ozone due to a lack of unsaturated moieties, OH radicals may play a more significant role by abstracting hydrogen from a wide range of organic species. Additionally, internal oxidant production via photosensitization—beyond multiphase uptake—could be worth mentioning, as it does not require gas-to-particle diffusion and may be particularly relevant under dry or cold conditions.

I appreciate the authors' efforts and look forward to seeing this work published in *Atmospheric Chemistry and Physics*!

**Reference**

Fleming, Lauren T, Peng Lin, James M Roberts, Vanessa Selimovic, Robert Yokelson, Julia Laskin, Alexander Laskin, and Sergey A Nizkorodov. 2020. 'Molecular composition and photochemical lifetimes of brown carbon chromophores in biomass burning organic aerosol', *Atmospheric Chemistry and Physics*, 20: 1105-29.

Gerrebos, Nealan GA, Julia Zaks, Florence KA Gregson, Max Walton-Raaby, Harrison Meeres, Ieva Zigg, Wesley F Zandberg, and Allan K Bertram. 2024. 'High Viscosity and Two Phases Observed over a Range of Relative Humidities in Biomass Burning Organic Aerosol from Canadian Wildfires', *Environmental Science & Technology*, 58: 21716-28.

Gregson, Florence KA, Nealan GA Gerrebos, Meredith Schervish, Sepehr Nikkho, Elijah G Schnitzler, Carley Schwartz, Christopher Carlsten, Jonathan PD Abbatt, Saeid Kamal, and Manabu Shiraiwa. 2023. 'Phase Behavior and Viscosity in

Biomass Burning Organic Aerosol and Climatic Impacts', *Environmental Science & Technology*, 57: 14548-57.

Huang, Ru-Jin, Lu Yang, Jincan Shen, Wei Yuan, Yuquan Gong, Haiyan Ni, Jing Duan, Jin Yan, Huabin Huang, and Qihua You. 2021. 'Chromophoric fingerprinting of brown carbon from residential biomass burning', *Environmental Science & Technology Letters*, 9: 102-11.

Knopf, Daniel A, Seanna M Forrester, and Jonathan H Slade. 2011. 'Heterogeneous oxidation kinetics of organic biomass burning aerosol surrogates by O 3, NO 2, N 2 O 5, and NO 3', *Physical Chemistry Chemical Physics*, 13: 21050-62.

Liang, Zhancong, Liyuan Zhou, Yuqing Chang, Yiming Qin, and Chak K. Chan. 2024. 'Biomass-burning organic aerosols as a pool of atmospheric reactive triplets to drive multiphase sulfate formation', *Proceedings of the National Academy of Sciences*, 121: e2416803121.

Vasilakopoulou, Christina N, Angeliki Matrali, Ksakousti Skyllakou, Maria Georgopoulou, Andreas Aktypis, Kalliopi Florou, Christos Kaltsonoudis, Evangelia Siouti, Evangelia Kostenidou, and Agata Błaziak. 2023. 'Rapid transformation of wildfire emissions to harmful background aerosol', *Npj Climate And Atmospheric Science*, 6: 218.

---

## Author Comment (AC2)

**Interactive comment on "Ozonolysis of primary biomass burning organic aerosol particles: Insights into reactivity and phase state"**

**Anonymous Referee #1**
**20 February 2025**

**General comments:**

Bogler et al. present an insightful study on the ozonolysis of biomass burning organic aerosols (BBOA) using flow reaction systems coupled with advanced mass spectrometry analysis. The manuscript thoroughly examines the chemical evolution of BBOA under varying relative humidity and ozone exposure conditions, offering a well-supported mechanistic discussion. This research addresses an important topic related to the atmospheric fate of wildfire and prescribed burning emissions, which play a crucial role in the atmosphere-climate system. The manuscript is well-structured, clearly written, and includes a detailed discussion of the results. Therefore, I believe it is well-suited for publication in *Atmospheric Chemistry and Physics* following minor revisions. Below, I provide a few specific comments that the authors may find helpful in further strengthening their study.

We thank the reviewer for their feedback and address the individual comments in the section below. The reviewer comments are in black, we address comments in blue and show modified sentences in red.

**Specific comments:**

Line 50: It may be worth noting that sunlight can also drive triplet-state chemistry and secondary oxidant formation inside BBOA particles, contributing to oxidative processes (Liang et al. 2024). Additionally, recent fieldwork by Vasilakopoulou et al. (2023) reported the rapid oxidation of biomass burning plumes, which could provide further context.

We have added the following section to the main text:

**Line 55-57:** Beyond conventional gas-phase oxidants, UV light can result in efficient formation of triplet-state chemistry and secondary oxidant formation within the aerosol (Bogler et al., 2021; Liang et al., 2024; Vasilakopoulou et al., 2023).

Line 75: Given the discussion on $O_3$ diffusion limitations in BBOA, it would be beneficial to elaborate on how liquid-liquid phase separation (LLPS) can lead to the formation of a highly viscous shell, which may further restrict oxidant penetration (Gerrebos et al. 2024; Gregson et al. 2023).

We have added the following text to the introduction:

**Line 79-81:** Recent work has also shown that BBOA can undergo a liquid-liquid phase separation, which could limit heterogeneous chemistry (Gerrebos et al., 2024; Gregson et al., 2023). These limitations could arise through differences in Henry's Law coefficient in the different layers, or through the development of a high viscosity shell.

Line 80: It would be helpful to provide additional context on the prevalence of these fuel types in ambient environments, particularly in wildfire-prone regions, to strengthen the relevance of the study.

We have added the following additional context:

**Lines 85-88:** These sources were chosen to represent major biomass burning emissions in Europe from residential wood stoves, and to elucidate changes associated with open burning major biomass burning sources in boreal forest regions.

Line 100: Since combustion conditions significantly influence BBOA composition, mentioning the typical burning temperatures in real wildfire and prescribed burn scenarios would enhance the discussion.

We added the following in the main text:

**Line 109-110:** Typical temperature conditions present during a wildfire can vary between 500 – 1500 $^{\circ}$C (Dennison et al., 2006).

Line 108: Is AADCO XX the brand of the gas generator?

Yes, we now updated the instrument specifications.

**Line 116-117:** …with an equal make up flow of clean air (clean air generator 737-250, AADCO instrument Inc.) to maintain a constant volume and pressure.

Line 160: A discussion on how the low solubility of certain BBOA components might affect the performance and sensitivity of the EESI-ToF analysis would be valuable.

Agreed, and we have added the following to Line 170. Note that we address this issue also in Section 3.1.2 (Line 222) regarding PAHs as a component of OA that is not soluble in water and not part of the analysis.

**Line 170-171:** Components of OA that are insoluble in water or those that do not effectively bind $Na^+$ are not detected, restricting the overall performance and sensitivity of the molecular analysis.

Line 240: It would be interesting to see whether the DBE plot captures the depletion of high DBE compounds upon oxidation, which could further support the loss of reactive species.

We agree that a DBE could capture the depletion of reactive species in an alternative way and could further show the ozonolysis reaction pathway of DB scission. However, for this study we have already focused the analysis on the key single species that are consistently detected and depleted which we have specifically highlighted through the analysis in the text.

Line 248: Knopf, Forrester, and Slade (2011) reported ozone uptake by LEVO particles, though the detailed chemical mechanism remains unclear. A brief mention of this could provide additional context.

We have added some context to the main text:

**Line 261-262:** In contrast, $f_{60}$ ($C_2H_4O_2^+$) remains largely stable throughout all $O_3$ exposures, which is not surprising because of the low uptake coefficient observed in previous studies (A. Knopf et al., 2011).

Line 282: The statement on volatility changes upon oxidation could be further elaborated, as increased volatility is often expected but depends on the specific reaction pathways involved.

We have added the following context:

**Line 301-302:** For instance, the ozonolysis of oleic acid results in carbon double bond scission and $C_9$ molecules (Gallimore et al., 2017).

Line 299: Were any aromatic compounds with vinyl side chains, such as coniferyl alcohol, detected? These species are commonly found in BBOA and are known to undergo ozonolysis (Fleming et al. 2020; Huang et al. 2021).

Although we do observe the molecular formula consistent with aromatics with vinyl side chains, we are reluctant to speculate along these lines because they are not the major species depleted during ozonolysis in our experiments, as we shown in Figure 3D, E, F.

Line 322: Gregson et al. (2023) suggested that the BBOA can undergo LLPS to form a highly-viscous shell.

We have added the following to the text:

**Line 342-343:** …but, Gregson et al., 2023 showed that a phase separated aerosol can have different diffusivity associated with the different layers, with the hydrophobic layer maintaining low diffusivity to high relative humidities (>90%).

Line 344: Did the author refer to 'mass accommodation' and 'bulk phase diffusion to reach mass equilibrium'?

Yes, in the discussion in what was lines 344-345 the bulk diffusivity of the molecules present in the BBOA increases, allowing for evaporation to proceed further than it would under dry conditions where diffusion is limited. We have altered the discussion for clarity.

**Line 365-366:** In addition to increasing the diffusion of $O_3$ into the BBOA particle, increased water content will increase the bulk diffusivity of the molecules present within BBOA allowing molecules to diffuse to the surface faster and evaporate out of the particle to achieve equilibrium between gas and condensed phase at wet conditions.

Line 391: The discussion on interfacial processes could be expanded, as the interfacial layer (~0.15 nm thick) in a 100 nm particle would only account for approximately 1% of the total particle volume. A quantitative perspective on this aspect would be helpful.

While it would be good to talk about these effects in a quantitative frame, but it would be difficult to perform this analysis. It would require constraints on the extent of oxidation (to connect O/C ratio changes to fraction of the BBOA changing) and evaporation occurring due to scission reactions occurring. Further, we would need a quantitative handle on the true carbon distribution in the aerosol to know how the O/C ratio is expected to change. With the extent of these unknowns and lack of constraints, we feel it wise to avoid overstepping our bounds here. For instance, having an ability to perform depth analysis or size-dependent density changes, which we do not have here, would provide constraints on the depth profile of this phenomena (Bell et al., 2017).

Line 400: The discussion on phase state as a key factor in multiphase oxidation is highly relevant. However, it would be useful to briefly comment on the oxidation of BBOA by other oxidants, such as OH radicals. Since BBOA has limited reactivity toward ozone due to a lack of unsaturated moieties, OH radicals may play a more significant role by abstracting hydrogen from a wide range of organic species. Additionally, internal oxidant production via photosensitization—beyond multiphase uptake—could be worth mentioning, as it does not require gas-to-particle diffusion and may be particularly relevant under dry or cold conditions.

We have added the following discussion on line 410 to highlight the works extension to other heterogeneous processes.

**Line 411-417:** These results should also extend to heterogeneous reactivity with OH or NO3 radicals. Although OH radicals will react with more species present within the aerosol than O3, the limited diffusivity observed with O3 should also limit OH radicals under similar dry and cold conditions. Further, given the surface reactivity of OH radicals they could also result in the build-up of a high viscosity medium at the surface, which would further exacerbate limited molecular diffusion. Of course the results here are focused on heterogeneous reactivity of gases at the interface with BBOA, and consequently the formation of radicals and oxidants within BBOA could be considered a more important source of BBOA oxidation than heterogeneous reactions (Bogler et al., 2021; Liang et al., 2024).

I appreciate the authors' efforts and look forward to seeing this work published in Atmospheric Chemistry and Physics!

**Reference**

Fleming, Lauren T, Peng Lin, James M Roberts, Vanessa Selimovic, Robert Yokelson, Julia Laskin, Alexander Laskin, and Sergey A Nizkorodov. 2020. 'Molecular composition and

photochemical lifetimes of brown carbon chromophores in biomass burning organic aerosol', Atmospheric Chemistry and Physics, 20: 1105-29.

Gerrebos, Nealan GA, Julia Zaks, Florence KA Gregson, Max Walton-Raaby, Harrison Meeres, Ieva Zigg, Wesley F Zandberg, and Allan K Bertram. 2024. 'High Viscosity and Two Phases Observed over a Range of Relative Humidities in Biomass Burning Organic Aerosol from Canadian Wildfires', Environmental Science & Technology, 58: 21716-28.

Gregson, Florence KA, Nealan GA Gerrebos, Meredith Schervish, Sepehr Nikkho, Elijah G Schnitzler, Carley Schwartz, Christopher Carlsten, Jonathan PD Abbatt, Saeid Kamal, and Manabu Shiraiwa. 2023. 'Phase Behavior and Viscosity in
Biomass Burning Organic Aerosol and Climatic Impacts', Environmental Science & Technology, 57: 14548-57.

Huang, Ru-Jin, Lu Yang, Jincan Shen, Wei Yuan, Yuquan Gong, Haiyan Ni, Jing Duan, Jin Yan, Huabin Huang, and Qihua You. 2021. 'Chromophoric fingerprinting of brown carbon from residential biomass burning', Environmental Science & Technology Letters, 9: 102-11.

Knopf, Daniel A, Seanna M Forrester, and Jonathan H Slade. 2011. 'Heterogeneous oxidation kinetics of organic biomass burning aerosol surrogates by O 3, NO 2, N 2 O 5, and NO 3', Physical Chemistry Chemical Physics, 13: 21050-62.

Liang, Zhancong, Liyuan Zhou, Yuqing Chang, Yiming Qin, and Chak K. Chan. 2024. 'Biomass-burning organic aerosols as a pool of atmospheric reactive triplets to drive multiphase sulfate formation', Proceedings of the National Academy of Sciences, 121: e2416803121.

Vasilakopoulou, Christina N, Angeliki Matrali, Ksakousti Skyllakou, Maria Georgopoulou, Andreas Aktypis, Kalliopi Florou, Christos Kaltsonoudis, Evangelia Siouti, Evangelia Kostenidou, and Agata Błaziak. 2023. 'Rapid transformation of wildfire emissions to harmful background aerosol', Npj Climate And Atmospheric Science, 6: 218.

**In addition to the references mentioned by the Reviewer, the following References were added to the manuscript:**

Bell, D.M., Imre, D., Martin, S. T., Zelenyuk., A.: The properties and behavior of α-pinene secondary organic aerosol particles exposed to ammonia under dry conditions, Physical Chemistry Chemical Physics 19.9 (2017): 6497-6507.

Bogler, S., Borduas-Dedekind, N., el Haddad, I., Bell, D., and Dällenbach, K.: How quality and quantity of brown carbon influence singlet oxygen production in aqueous organic aerosols, EGU General Assembly Conference Abstracts, ADS Bibcode: 2021EGUGA..2310743B, EGU21-10743, https://doi.org/10.5194/egusphere-egu21-10743, 2021.

Dennison, P. E., Charoensiri, K., Roberts, D. A., Peterson, S. H., and Green, R. O.: Wildfire temperature and land cover modeling using hyperspectral data, Remote Sensing of Environment, 100, 212–222, 2006.

Gallimore, P. J., Griffiths, P. T., Pope, F. D., Reid, J. P., and Kalberer, M.: Comprehensive modeling study of ozonolysis of oleic acid aerosol based on real-time, online measurements of

aerosol composition, Journal of Geophysical Research: Atmospheres, 122, 4364–4377, https://doi.org/10.1002/2016JD026221, 2017.

**Anonymous Referee #2**
**21 March 2025**

The manuscript presents a detailed investigation into the aging of biomass burning organic aerosol (BBOA) by ozone, with a focus on how the chemical compositions change across different fuel types and ozone exposure conditions. They also illustrate how humidity affects reactivity, offering useful insights into the underlying processes.

The manuscript is well-written, has a solid experimental design, and clearly presents results that have broader impacts. However, some experimental details require further clarification to strengthen the manuscript for publication. Below are my comments:
We thank the reviewer for their feedback and address the suggestions individually below. The reviewer comments are in black, we address comments in blue and show modified sentences in red.

L85. Were any light sources used in the experiments, or were they conducted entirely in dark?
As soon as the particles enter the sampling lines, they are kept in the dark as the particle sampling lines, holding chamber and oxidative flow reactor (details in Li et al., 2019) are made out of stainless steel, which is not transparent. Only the smog chamber is transparent. However, the chamber experiments were carried out over night without exposing the particles to any daylight, along with visible lights being off during the duration of the experiment. Therefore we rule out the influence of photolytic reactions on the presented results.

L92. In Fig S1-S2, certain labels (for example, "Dekati") appear in the experiment setup schematics but are not referenced or explained in the main manuscript or SI. Could the authors please clarify these labels?
Thank you for the suggestion. We updated Figure S1, added two references to the caption of Figure S2 and improved the ozonolysis procedure description regarding the Dekati (ejector dilutor).
**Line 111-112:** 50 – 150 μg m-3 in the sampling lines after dilution with an ejector dilutor, DI-1000, Dekati Ltd.).

L95. In Table 1, all of the pine experiments were carried out in open fire. Is there a particular reason for that?
The pine wood used in the experiments includes branches with needles, and its combustion under open fire conditions is meant to quasi mimic the conditions of a wildfire. In contrast, the spruce and beech wood processed into logs are burned in a stove, simulating the conditions of residential wood burning for heating purposes, for which this type of pine wood would not be used.

I suggest labeling experiments 1-3 as spruce1, open1, and beech1. I got confused between the experiment labels and the fuel types a few times while reading the paper.
Thank you for the suggestion, we updated the labelling in tables, text and figures accordingly, to improve the readability.

L108. What does "AADCO XX" mean?

Thank you for pointing out this unclear instrument description. AADCO XX referred to the clean air generator used to supply the flow of clean air into the holding chamber making up for the sampling flow. We now specified the instrument description.

**Line 116-117:** …with an equal make up flow of clean air (clean air generator 737-250, AADCO instrument Inc.) to maintain a constant volume and pressure.

L128. The O3 concentration steps are presented in Table S2, but the "step#" column is confusing. The authors should clarify or relabel these steps for better understanding. Additionally, all parameters in the table should include units.

To improve the understanding of the Tables S1-S2, we added to the caption and included the unit information *[L/min]* for each column, which was previously summarized in a joint cell for all columns specifying flow conditions.

**Addition to caption of Table S1:** The column "step #" numbers the varying ozone conditions set for each experiment; the label "(POA)" denotes steps where the ozone concentration was set to 0 to measure the composition of POA (see the method section of the main manuscript for more details).

**Addition to caption of Table S2:** The column "step #" numbers the varying ozone and RH conditions set for each experiment; the label "(POA)" denotes steps where the ozone concentration was set to 0 to measure the composition of POA (see the method section of the main manuscript for more details).

L185. From what I can tell, an exponential fit might work better for the three open fire experiments.

Thank you for reviewing the supplementary information thoroughly. We reviewed the fits but decided to keep the linear fit for the open fire experiments. Exponential fits were only used for experiments where additional particle instruments (data is not used in this study) required higher flow demands from the sampling chamber, which was not the case for the open fire experiments.

L191. In Fig 1a, the label CHOgt1 is not defined.

Thank you for spotting this missing explanation, which we now added to the Figure's caption.

**Caption Fig.1:** …In (a), the colours differentiate the groups of ions containing zero (CH), one (CHO1) or more than one O atom (CHOgt1)….

L199. The authors introduced the tracer f60 but did not define it.

To improve the understanding of these species labels, we added a statement to clearly link m/z 60, i.e. the ion C2H4O2+, to the fraction 60, abbreviated as $f_{60}$.

**Line 208-211:** We also consistently found a clear signal up to 2.7 % for the marker species of primary emitted BBOA at m/z 60, $C_2H_4O_2^+$. This ion is resulting from the pyrolysis of anhydrous sugars like levoglucosan and is commonly labelled as $f_{60}$ (Aiken et al., 2009; Cubison et al., 2011; Lee et al., 2010; Simoneit et al., 1999). Though we do observe variations in relative fractions of $f_{60}$ up to 0.8% between experiments (Figure 2c-e), these are likely based on variations in the fuel material and burning conditions and are within the variability for complex burning experiments.

L223. Since the pine experiments were conducted under different combustion conditions, do the authors think this could affect the results? Does the conclusion regarding fuel type still hold?

Of course the combustion conditions will impact the results. We observe in Figure 3 that the composition of the BBOA varies from open burning (pine) to residential burning (spruce). We have included different combustion sources to demonstrate the robustness of the data and to show that regardless of the burning type that we have investigated we observe similar phenomena observed in Figures 2-4. We have modified the text accordingly:

**Line 233-235:** The increase in O/C with $O_3$ exposure is reproducible across all experiments performed and thus the O/C ratio is useful as a characteristic bulk feature for each fuel type, and highlights the consistency of this process regardless of the combustion source (residential stove vs. open burning).

L243. In Fig 2c-e, could you label the markers more clearly? It's hard to tell which marker represents which experiment.

Thank you for the suggestion. We updated the legend entries for Fig 2c-e to focus on the different markers, i.e. experiments. The meaning of markers with no fillings was added to the caption.

**Addition to caption of Fig.2:** …Empty markers represent POA conditions of the respective experiment.

L243. Now that f44 is defined here, I assume f60 is defined similarly. Although these tracers have been used in other studies (Cubison et al., 2011), I suggest that the authors briefly explain what f44 and f60 represent and why these tracers are picked. Providing this clarification would help readers, especially non-experimentalists unfamiliar with these terms, better understand the f44 vs f60 space plots presented here.

Thank you for this suggestion. We are happy to add some introductory sentences on why f44 vs. f60 plots are useful and thereby improve the overall understandability of these results for a broader audience.

**Line 255:** Fig. 2c-e show scatter plots of f44 vs. f60, which is an established method to visualize the evolution of oxidative aging in BBOA: f44 is the mass fraction of m/z 44 which is mostly CO2+, i.e. the end of an oxidative chain. The fraction at m/z 60, f60, is used as a marker species for primary BBOA (Section 3.1.1). Upon oxidation, it is expected that f44 increases and f60 decreases or stays the same. Indeed, Fig. 2c-e show an increase in f44, thereby illustrating an overall increase in the oxidative state of the particles upon O3 exposure.

L271. I would suggest the authors to add figure numbers here when you talked about the comparison.

Thank you for the suggestion, we added references to Fig. 3a-c to the comparison statements.

L275. I was wondering if the authors have considered whether the fuel types or combustion methods could influence the conclusions here. It would be beneficial to address how these factors might impact the results and whether the findings can be generalized across different conditions.

We added on the following lines to address this point in Lines 287-296, and in the conclusions section Lines 399-400:

**Line 289-291:** This observations points at parameters other than the O3 exposure (Section 3.33) determining the progression of the ozonolysis; and this limitation starts at an O3

exposure threshold below the maximum exposures tested in the experiments here regardless of the fuel type.

**Line 297-299**: Therefore, the different fuel types have different emissions that react more or less with O3 (beech emitting species that react less and spruce/pine emitting species that react more).

**Line 400-401**: for all combustion sources investigated here.

L320. I appreciate the mechanistic insights presented here; however, they do not fully explain all the observations.
We believe that this is in reference to the reviewers point on L339, and will address the reviewer's comment there. We apologize if we do not answer the question exactly, since we did not exactly know what the reviewer was referring to.

L339. The manuscript does not explain why the remaining fraction of these compounds is lower at 50% relative humidity compared to 80%. Could the authors provide an explanation or potential mechanistic insights for this?
We attribute the lower fraction at 50% relative to 80% to come from experimental variability. Note in Figure 4, the variability between $50 - 200$ ppb hr$^{-1}$ from individual experiments can vary between $0.1 - 0.2$. Similarly the difference 50% and 80% on Figure 6 are on a similar order, and we attribute these differences to experimental variability.

We made a note of this in the text at **Lines 360-362:** Note there is some experimental variability in the extent of loss shown in Figure 6 between 50% and 80%, where the 50% data point is occasionally lower than the 80% data point. These differences are on a similar order to the variability shown in Figure 4. Overall,

**Citation**: https://doi.org/10.5194/egusphere-2025-385-RC2